# Vertical Mental Timeline Is Not Influenced by VisuoSpatial Processing

**DOI:** 10.3390/brainsci14020184

**Published:** 2024-02-17

**Authors:** Alessia Beracci, Marco Fabbri

**Affiliations:** 1Department of Psychology Renzo Canestrari, University of Bologna, 40126 Bologna, Italy; alessia.beracci@unibo.it; 2Department of Psychology, University of Campania Luigi Vanvitelli, 81100 Caserta, Italy

**Keywords:** mental timeline, vertical space, STEARC effect, visuospatial processing, temporal decision task, Time-to-Position task, more-is-up metaphor

## Abstract

The results examining the direction (bottom-to-top vs. top-to-bottom) of the mental vertical timeline are not conclusive. The visuospatial processing of temporal stimuli along vertical space could influence this time representation. This study aimed to investigate whether and how the visuospatial processing stage modulated the vertical timeline in an online temporal categorization task. In three studies, Italian university students (*N* = 150) responded more quickly to words expressing the past with a down arrow key, and more quickly to words expressing the future with an up arrow key, irrespective of whether the words were located in the top, middle, or bottom space (Experiment 1), or were presented downward (from top to bottom; Experiment 2A) or upward (from bottom to top Experiment 2B). These results suggest that the representation of time was not influenced by the visuospatial processing. The daily experience with verticality (e.g., to reach the attic, the lift goes up) could explain the bottom-to-top direction of the mental timeline.

## 1. Introduction

In cognitive science, interactions between space, time, and numbers have been extensively studied, attempting to understand the cognitive representations of these concepts [1,2,3]. The representation of both time and numbers might be rooted in cortical networks which are also involved in spatial cognition, with a particular involvement of the parietal cortex [3,4,5]. In addition to the SNARC (Spatial-Numerical Association of Response Codes [6]) effect, there is evidence of an STEARC (Spatial-TEmporal Association of Response Codes) effect [7,8]. This STEARC effect reflects faster response latencies when short/early/past time is associated with left space/key and when long/later/future time is associated with right space/key, suggesting a mental timeline [7,8,9]. The STEARC effect has been replicated using different types of temporal stimuli and experimental manipulations [10,11,12,13,14,15,16,17,18,19,20,21]. In many studies, lateralized manual keys along the horizontal axis provided the spatial information [2] and thus, the STEARC effect reveals an interaction between time and associated motor responses. However, a space–time congruency effect has also been proposed [2,9], and this congruency effect reflects quicker processing of stimuli when they have congruent characteristics across different dimensions. An example of the space–time congruency effect is provided by long durations or future expressions presented in the right hemifield, without any influence of response selection. This congruency effect could claim for the involvement of visuospatial processing on temporal processing or mapping. Torralbo et al. [22] presented pictures of human silhouettes with a time-related word either at the left or right of the silhouette’s face and the authors found that participants were quicker to process time when there was a congruency between past time and left space as well as between future time and right space. In a similar way, Santiago et al. [9] and Fabbri et al. [23] found both a space–time congruency effect (i.e., past expression or short duration positioned on the left and future expression or long duration positioned on the right) and an STEARC effect (i.e., past/short stimuli judged with the left key and future/long stimuli judged with the right key). Using a visuospatial priming paradigm, Di Bono et al. [24] found that leftward cues induced an underestimation of temporal duration and rightward cues induced overestimation, in a similar way to the study performed by Vicario et al. [25], who showed that the judgments of temporal duration were influenced by the spatial position of stimuli. These data might suggest that the experience determined by reading and writing habits in a specific culture can modulate the STEARC effect (but see [26,27,28] for contrasting results). Indeed, the rightward (or leftward for Arabic culture) movement of the eye during the reading and writing experience modulates the chronological order in which the left and right sides of space are experienced along a left-to-right direction [19,29]. In line with this assumption, it has been reported that eye movements reflect mental looking-through-time during language comprehension about time [30].

### 1.1. Vertical Timeline

Beyond a horizontal timeline, a vertical STEARC effect has also been found [30,31,32,33,34,35,36,37,38,39,40,41,42], but see [7,15,43] for contrasting results. On one hand, some studies have indicated a bottom-to-top direction of the timeline [30,31,32,33,34,35,37,38,39,41], whereas other studies have reported a top-to-bottom direction [36,40,42], indicating no conclusive assumptions for the directionality of the vertical timeline. These mixed results could be due to several methodological aspects. First, different types of temporal stimuli have been used in these studies. When a vertical STEARC effect was not found, past-related (e.g., World War II) and future-related (e.g., holidays on Mars) verbal stimuli [15], pictures of buildings and cities (e.g., ancient ruins for the past and science fiction scenes for the future), or modified pictures of famous actors (e.g., a picture of Brad Pitt when he was young, is an adult, and will be old) [40] were used. When a vertical STEARC effect was found, temporal stimuli recalling a mental temporal progression rather than fixed moments in time were used. Moreover, the temporal task involved before/after or earlier/later comparisons (e.g., March comes earlier than April (e.g., refs. [12,31,35,36])), the detection of an order of progression (e.g., cards depicting an apple being eaten (e.g., refs. [20,33])), or shorter/longer estimations (e.g., target stimuli with variable durations in the millisecond or second range (e.g., ref. [32])). A second methodological aspect regards the number of stimuli presented during the experiment. Indeed, for example, Miles et al. [40] in their first experiment presented 22 photos (11 for the past and 11 for the future) twice within each block (for a total of four blocks, corresponding to 176 trials), whereas in their second experiment, they presented three photos of Brad Pitt and Jet Li (for a similar procedure, see Tversky et al. [21] with cards for breakfast, lunch, and dinner). In a similar way, Woodin and Winter [43] requested participants to place onto a piece of A4 paper (with or without central axes) the time words “past”, “future”, “earlier”, “earliest”, “later”, and “latest”, while Leone et al. [39] requested participants to freely represent on a PC screen different temporal events, such as zones of time (past, present, and future), seasons, days of the week, parts of the day, and timelines of personal or historical events. On the other hand, Beracci et al. [31] used 20 selected temporal words (see Beracci et al. [31] for the selection procedure of these stimuli) repeated nine times in two different blocks (for a total of 360 trials; see also Dalmaso et al. [36] with 224 or 272 trials in their experiments) in a binary task. Using a “no binary task” in a similar way to the position task used by Woodin and Winter [43] or by Leone et al. [39], Beracci et al. [31] presented 20 temporal words (10 referred to the past and 10 referred to the future), suggesting that the number of trials has an impact on the direction of the timeline in binary and nonbinary tasks. Although there is an additional methodological issue related to the response modality adopted in these studies (e.g., see Beracci et al. [31] for a review of this issue), we decided to analyze another methodological aspect which is related to the implicit [36,42] and explicit (e.g., ref. [36]) nature of the task. Indeed, Topić et al. [42] required participants to detect a target (e.g., an orange) within a 7 × 7 search array. Specifically, the authors designed a go/no-go task, in which a target object was presented for a variable duration (300, 600, 1200, or 1500 ms) in several spatial positions (vertically, horizontally, and diagonally) of the array and participants had to detect it (go trial) by pressing a central button, or refrain from pressing the key (no-go trial) when the target was absent. Interestingly, Topić et al. [42] found a spatial–temporal association along the vertical and left-diagonal (from top to bottom) axes. When an explicit task was adopted, generally, two response keys were adopted. These keys were either literally placed vertically (e.g., refs. [31,36,40]) or reminded (or better provided) the verticality (e.g., refs. [7,31,32]). Thus, the presence of lateralized keys could induce a specific orientation of the timeline. However, Dalmaso et al. [36] did not report any vertical STEARC effect when an implicit task was performed, while the authors found an STEARC effect in explicit tasks. In the present study, we tried to consider several methodological aspects in order to disentangle these discrepancies in the literature.

In addition to the above methodological aspects, another possible factor could be related to a different visuospatial processing experience. Indeed, the bottom-to-top orientation of the mental timeline could be based on vertical cases experienced by people in their daily life. For example, when we fill a glass of water, the level of water goes up; to reach the last floor of a building, the lift goes up from the ground floor; or a plant grows upward in its growth process. In line with the framework of the “more-is-up” metaphor (ref. [44]; see also Tropic Embodied and Situated Theory of cognition or TEST model [45]), the conceptualization of abstract concepts along the vertical space is based on concrete sensorimotor experience of the physical world. A possible example could be the experience of reading daily emails, given that a new email (i.e., future) appears at the top of the Outlook mailbox, while the oldest emails (i.e., past) are stored at the bottom of the mailbox. On the opposite hand, the top-to-bottom direction of mental time representation could be grounded in repeated experience with physical laws [46], such as the law of gravity which determines the falling of objects [42]. In addition, the possible downward direction could derive from the experience of reading and writing habits [47]. In Western culture, people read and write from the top of the page toward the bottom of the space (e.g., to read and/or write an article on a computer screen as well as to scroll downward to read new text messages on a mobile phone), and this concrete sensorimotor experience (i.e., the eyes move downward) could support the CORrelations in Experience (CORE) principle in which conceptions of time are shaped by cultural practices [19]. To the best of our knowledge, only two studies have addressed the visuospatial processing in the space–time congruency effect. First, Casasanto and Bottini [48] required participants to categorize temporal expressions which were rotated downward or upward orthographically. The idea was that the experience of reading text with downward or upward rotation should influence the direction of the timeline. Thus, participants were requested to read stimuli and instructions rotated 90° clockwise or 90° counterclockwise. Participants were requested to press two keys to indicate whether a word referred to the past or future, and in this case the keys were oriented vertically, determining up and down buttons. The authors found that participants reported a bottom-to-top or a top-to-bottom vertical STEARC effect as a function of the exposed downward or upward stimulation: past-top and future-bottom associations with a downward orthography and past-bottom and future-top associations with an upward orthography. Second, Beracci et al. [32] manipulated the spatial location of temporal reference or target stimuli along the vertical axis (i.e., at the top, center, or bottom) and asked participants to perform a time estimation task. The authors only found an STEARC effect (i.e., past-bottom and future-top associations) demonstrated by the interaction between the temporal durations of the targets and spatial position of the keys, independently from the spatial positions of the stimuli (either reference or target). These mixed results could be related to the different temporal stimuli used in both studies (temporal expressions vs. temporal durations, respectively), limiting any comparison between studies. Thus, the effect of the visuospatial processing stage on the space–time association along the vertical axis deserves further study.

### 1.2. Predictions of the Present Study

The main goal of this study was to investigate the possible interaction between the STEARC effect and visuospatial processing along the vertical space. This interaction should emerge when there is a congruency-like effect between STEARC and visuospatial processing. Specifically, a triple interaction between time-related words, their spatial (vertical) position along the screen, and the spatial position of arrow keys should be found. For example, past words placed at the bottom of the screen pressed by the bottom arrow key (for a bottom-to-top direction of the timeline) or past words placed at the top of the screen pressed by the top arrow key (for a top-to-bottom direction of the timeline) could describe this triple interaction. The expectation of this triple interaction could be supported by neuroimaging evidence of the involvement of the right posterior parietal cortex (PPC), especially the posterior part along the intraparietal sulcus (IPS), for a generalized magnitude system for space, time, numbers, and other magnitudes [3,4,5].

However, STEARC and visuospatial processing effects could be found without any interaction between them. In this case, two possible scenarios could be observed: (a) both effects have the same direction, either bottom-to-top (e.g., past words categorized more quickly with the down arrow key and past words categorized more quickly when they appear at the bottom) or top-to-bottom (e.g., past words categorized more quickly with the up arrow and past words categorized more quickly when they are placed at the top); (b) the effects have different directions (e.g., past words pressed more quickly with the down arrow and future words pressed more quickly with the up arrow for a bottom-to-top STEARC effect vs. past stimuli placed at the top and future words placed at the bottom, categorized more quickly than the reversed spatial presentation of the stimuli). The first scenario could advance the idea that both visuospatial processing (i.e., perceptual) and motor processing stages are involved in the vertical representation of time. Although the second scenario seems to be counterintuitive, this possible “dissociation” between visuospatial and motor processing stages could be in line with the intermediate-coding account proposed by Kong and You [49]. This account states that the temporal meanings activate a spatial code, such as left and right (i.e., here the spatial code is bottom or top). The association between temporal expressions and spatial codes is preexisting in the long-term memory. The crucial point of the intermediate-coding account is that the spatial codes could be associated with response (i.e., here, down and up arrows) and physical space codes (i.e., top and bottom spaces), according to task characteristics. Thus, the STEARC effect should derive from the spatial information provided by the bottom and up arrows [2], in line with the polarity correspondence account [50], while the visuospatial processing effect should derive from the spatial physical codes, in line with the CORE account [19], according to the law of gravity. In addition, Riemer et al. [51] demonstrated that the right PPC inhibition along the IPS affected the space–time congruency effect (i.e., visuospatial effect) but not the STEARC effect, suggesting that different neuronal mechanisms were related to the STEARC (i.e., motor) and congruency (i.e., visuospatial) effects. In line with this assumption, for example, Vallesi et al. [52] have shown that the spatial representation of time produced a response code that preactivated the motor cortex with left-short and right-long mapping, whereas the space–time congruency effect seemed to be more involved in the parietal areas [51].

Finally, two other possibilities should be acknowledged: (1) to find an STEARC effect only; (2) to find a visuospatial processing effect only. These two possibilities should be revealed by either an interaction between temporal stimuli and spatial response keys (outcome n.1) or an interaction between temporal words and the spatial positions of the stimuli (outcome n.2). In both cases, a possible idea at which stage the spatial–temporal association should arise, in a similar way to what has been reported for numbers (e.g., ref. [53]), should be advanced.

### 1.3. The Present Study

In our Experiment 1, participants were requested to categorize 20 Italian words (see Appendix B) referring to the past or future. These target stimuli were located at different positions on the screen (bottom, center, and top spaces), with the intent to replicate and extend the experimental procedure adopted by Beracci et al. [32] using different temporal stimuli (here temporal words vs. the durations in Beracci et al. [32]). The vertical information was also provided by pressing the up or down arrows of a standard keyboard as response keys. The choice to use the directional arrows was justified by several aspects. First, we carried out online data collection due to the COVID-19 lockdown. Consequently, we selected the spatial arrows in order to simplify the experiment’s administration because they are present on all standard keyboards. Second, Beracci and Fabbri [31] showed a similar STEARC effect using either up-down arrows on a keyboard (i.e., analogous concept of verticality) or (physically) up-down keys on an external response box positioned vertically (i.e., literal concept of verticality). Third, the directional arrows are commonly used to move the cursor along the vertical direction of the screen, such as to scroll down a word text or a pdf file, and they could “remind” the experience of reading along the vertical dimension. Finally, they could be pressed with right and left fingers, given that Beracci et al. [32] have already controlled for the role of whichever hand is used to respond in influencing the STEARC effect. Consequently, the directional arrows could resemble more a vertical than a radial dimension, in line with previous studies which used the same keys or keys arranged vertically on the keyboard [7,31,32,54,55,56]. In our Experiment 2, we added a further element of visuospatial processing, following the manipulation proposed by Casasanto and Bottini [48]. The same temporal words used in Experiment 1 were arranged in a downward direction (from top to bottom in Experiment 2A), and in an upward direction (from bottom to top in Experiment 2B), located in three spatial positions along the screen, as in Experiment 1.

In all experiments, after the temporal categorization task, participants were asked to perform a Time-to-Position task. Participants had to place all temporal words along a vertical line (see Beracci et al. [27] for the same task along the horizontal dimension). Specifically, participants were presented with a series of vertical lines (a single line for each temporal stimulus presented) and they were requested to indicate the spatial position along the line of each temporal word. In our opinion, the Time-to-Position task was important because it used a continuous response format and not a speeded binary choice as in the temporal categorization task. In this way, participants could express their own time representation vertically. Thus, this task could confirm and complete the results provided by bimanual speeded responses (see [27,31,39,41,43] for a similar procedure). For this task, two possible data patterns should be expected: (1) all past words are placed on the bottom (or on the top) side of the line and all future words are placed on the top (or on the bottom) side of the line, suggesting that the vertical representation of time is mainly determined by task setting (e.g., vertical keys or vertical lines); (2) the temporal words are placed along the line with a specific order (upward or downward), suggesting that participants are probably induced to “travel” along the vertical line to perform this task. The presence of the visual line could not necessarily determine a specific order of temporal words along the line, given that we used words referring to past and future without any specific temporal order. Indeed, the temporal words are usually used to refer to different time periods, which are not a priori ordered. For example, the words “after” and “soon” could be both used for referring to a time span of 5 or 10 min, as well as 1 h. Thus, whether participants place “after” and “soon” in order should indicate that there is a “temporal difference” between them (i.e., they refer to different time periods).

## 2. Experiment 1

### 2.1. Materials and Methods

We recruited a group of 61 Italian students (53 females; mean age = 24.31 years; SD = 3.15 years) for participation in an online study, in exchange for course credits. The sample size estimation was made on the basis of a statistical power analysis with G*Power 3.1. [57], performed on data from a meta-analysis on the space–time congruency effect [2]. The effect size in this study was Cohen’s *∫* = 0.23, which reflected a medium size according to Cohen’s criteria [58]. Fixing alpha at 0.05 and beta at 0.80, the needed sample size was approximately equal to 40 for within-group comparison. Therefore, our sample size kept Type I errors at the desired level of 5% with a concomitant reduction of any Type II errors. Based on the Edinburgh Handedness Inventory (EHI) score [59], 53 participants were right-handed (M = 85.00; SD = 17.86) and 8 were left-handed (M = −68.00; SD = 29.97). Data from 5 participants (4 females and 1 male) were not included because they reported more than 25% of errors (M = 46.06%, SD = 7.99%; see Grasso et al. [28] for similar exclusion criteria of outliers). Thus, the data of the remaining 56 participants (49 females; mean age = 24.41 years; SD = 3.26 years; 50 right-handed) were analyzed. All participants had normal or corrected-to-normal vision and were not informed of the purpose of the study. Informed consent was obtained from all individual participants included and the study was approved by the Ethical Committee of the Department of Psychology at the University of Campania Luigi Vanvitelli.

We carried out an online data collection because of COVID-19 lockdown in Italy, and social restrictions prohibited access to universities and laboratories. The online data collection was performed using the online version of the E-Prime 3.0 software [60]. First, each participant had to download the experiment folder in their computer, excluding any influence of participant’s internet speed on the recording of reaction times (RTs). We chose to exclude mobile devices, such as phones and tablets, to avoid unreliable RT measurements. All stimuli were presented in the browser window and responses were measured using the down and up arrow keys of a standard keyboard. The stimuli consisted of 20 Italian temporal words (Appendix B), 10 referring to the past (e.g., “*ieri*”, English translation “yesterday”) and 10 referring to the future (e.g., “*domani*”, English translation “tomorrow”), that appeared in three different positions of the screen: at the top, in the center, and at the bottom. We designed the experiment using the E-Prime 3.0 “*General tab*” which allowed the specification of the target to be displayed as well as the observable properties of that object. Thus, each stimulus position corresponded to the BOTTOM (i.e., corresponding to 75% of the vertical length of the display size), CENTER (i.e., corresponding to 50% of the vertical length of the display size), and TOP (i.e., corresponding to 25% of the vertical length of the display size) position of the ALIGHVERTICAL field in the E-Prime program. In this way, the vertical alignment of the stimuli was determined within the defined display area. In addition, we selected in E-Prime 3.0 the option to match desktop resolution at runtime in the DISPLAY feature, and thus, the spatial position of target was kept constant independently from the screen size of the personal computer used by participants to perform the tasks. Before the presentation of the temporal words, a white fixation plus (+) symbol was presented.

Participants were required to categorize the temporal words and were instructed to press the down and up arrow keys with the left and right index, respectively (see Beracci et al. [31,32] for similar methodological aspects). The fixation symbol appeared at the center of the screen for 1000 ms. After the fixation point, a temporal word (the target stimulus) was presented in different locations. In all trials, the target stimulus lasted 5000 ms or until an individual response had been recorded (Figure 1). The task was performed twice, in two different blocks, where the instruction-key assignments were counterbalanced. In one block, the down key was pressed with the left index to categorize the past words and the up key was pressed with the right index to categorize the future words. In the other block, the down arrow was pressed with the left index to categorize the future words and the up arrow was pressed with the right index to categorize the past words. The order of blocks was counterbalanced within participants. In each block, 180 trials were presented in a random order, and in total 360 trials were presented. Before the test, a training session was run with 10 trials. The experimental procedure lasted approximately 20 min.

At the end of the temporal categorization task, participants were requested to perform an online version of the Time-to-Position task using the E-Prime 3.0 software. Participants placed all of the 20 temporal words (Appendix B) along a 10 cm-long line flanked by two labels, “*passato lontano*” (“distant past” in English) on the bottom side of the line, and “*futuro lontano*” (“distant future” in English) on the top side of the line. Each line was positioned centrally on the screen. To place each word, participants had to move (downward or upward) a circular cursor positioned centrally on the line, using a mouse. This cursor appeared on the center of the line at the beginning of each trial. At the beginning of each trial, participants read the instruction “The word to position on the line is …” (originally “*La parola da positionare sulla linea è …*”) in order to show by turns the target word to position (Figure 2). Each target word was randomly presented.

For the temporal categorization task, the mean reaction times (RTs) of correct responses were calculated using the software SPSS version 20. To improve the internal validity of the study, RTs above or below 3 SDs of the mean were discarded as outliers (in total, we discarded about 9% considering both response errors and outliers). A three-way repeated measures ANOVA was carried out on RTs and the absolute Number of Errors (NEs, defined by the instruction of the blocks, e.g., to press the down arrow for past words), with Temporal Word (past vs. future), Response Position (down key vs. up key) and Stimulus Location (bottom, center, and top) as within-subjects factors. When a reliable significance was found, the Bonferroni post hoc test was run. We set the alpha level at 0.05 and we reported the effect size (as eta partial square *η*^2^*_p_*).

For the Time-to-Position task, we calculated the Euclidian distance (in mm) from the bottom of the line to the subjective mark positioned on the line for each participant and for each temporal word. Then, for each participant we calculated which individual order of temporal words was provided and, finally, we tested whether a linear model (*R*^2^) fitted well with the subjective order of the temporal words.

### 2.2. Results

For the temporal categorization task, all descriptive data for RTs and NEs (with their relative SD) are displayed in Table 1.

The Temporal Word × Response Position × Stimulus Location ANOVA on RTs revealed a significant main effect for Stimulus Location (*F*(2,55) = 37.44, *p* < 0.001, *η*^2^*_p_* = 0.40), with faster responses for stimuli presented at the center (M = 1093 ms, SD = 282 ms) rather than the bottom (M = 1161 ms, SD = 281 ms, *p* < 0.001) or top part of the screen (M = 1163 ms, SD = 285 ms, with *p* < 0.001). We found a significant Temporal Word × Response Position interaction (*F*(1,55) = 22.31, *p* < 0.001, *η*^2^*_p_* = 0.29), suggesting an STEARC effect: faster responses with the down key for past words (M = 1081 ms, SD = 269 ms) than with the up key (M = 1196 ms, SD = 293 ms) with *p* < 0.001, and faster responses for future words with the up key (M = 1070 ms, SD = 269 ms) than with the down key (M = 1211 ms, SD = 302 ms), with *p* < 0.001. No other significant main effects and interactions were found (all *F*s < 2.88 with *p*s > 0.06, *η*^2^*_p_* < 0.05).

The same ANOVA on NEs showed a significant interaction between Temporal Word and Response Position (*F*(1,55) = 7.81, *p* < 0.001, *η*^2^*_p_* = 0.12), confirming the previous STEARC effect: participants made fewer errors for the past words with the down key (M = 4.36, SD = 4.56) than with the up key (M = 5.73, SD = 5.77), with *p* < 0.05, and they made fewer errors for future words with the up key (M = 5.26, SD = 6.17) than with the down key (M = 6.57, SD = 7.15), with *p* < 0.05. Also, significant interactions between Response Position and Stimulus Location (*F*(2,55) = 8.57, *p* < 0.001, *η*^2^*_p_* = 0.13) and between Temporal Word and Stimulus Location (*F*(2,55) = 10.66, *p* < 0.001, *η*^2^*_p_* = 0.15) were found. For the former interaction, the post hoc test did not reveal any significant comparisons for the down key, although there were fewer NEs when stimuli were positioned at the bottom (M = 5.09, SD = 5.94) rather than at the center (M = 5.33, SD = 5.51) or at the top of the screen (M = 5.98, SD = 6.11). When the up key was pressed, the stimuli located at the top (M = 4.67, SD = 4.90) induced fewer NEs than those when stimuli were located at the bottom (M = 6.93, SD = 7.74), with *p* < 0.05. At the same time, the stimuli presented at the bottom of the screen induced higher NEs than those presented at the center of the screen (M = 4.88, SD = 5.26), with *p* < 0.05. As regards the Temporal Word × Stimulus Location interaction, the post hoc test revealed fewer NEs for future words presented at the top (M = 5.18, SD = 5.64) and center (M = 5.15, SD = 5.33) than at the bottom of the screen (M = 7.41, SD = 9.00), with *p* < 0.05 and *p* < 0.001, respectively. Although participants made fewer errors for past-related stimuli located at the bottom (M = 4.61, SD = 4.68) with respect to those located in the other two spatial positions (center: M = 5.06, SD = 5.44; top: M = 5.48, SD = 5.37), no significant comparisons were found. No other main effects and interactions were shown (all *F*s < 2.67 with *p*s > 0.07, *η*^2^*_p_* < 0.05).

As shown in Appendix B, Italian participants placed the 20 temporal words in an ordered pattern from bottom to top along a linear timeline and this order fitted well with a linear model (linear fit model: *R*^2^ = 0.798, SD = 0.13, *t*(55) = 44.24, *p* < 0.001). Thus, we decided to further test a vertical STEARC effect in line with previous studies [27,31]. A regression analysis for the repeated measures design was performed [61]. Specifically, we imposed on all participants the order of temporal words according to the results obtained in the Time-to-Position task, and then we calculated the RT differences between the up key and down key for each temporal word, according to this word order. Then, we calculated the linear regression coefficient predicting these values depending on the spatial position of the responding keys, so that a negative regression coefficient denoted a relatively higher advantage for the past words when using the down (compared to the up) key and vice versa for the future words, suggesting the STEARC effect. The slope coefficient was tested against zero [60]. We obtained a negative slope coefficient (b = −19.46, SD = 29.02), which significantly differed from zero (*t*(55) = −5.02, *p* < 0.0001, *Cohen’s d* = −0.95). In the Appendix A, a graphical representation of the regression analysis is provided.

In order to test possible individual differences in our results, first of all, we counted how many participants deviated from the order found in the Time-to-Position task (Appendix B), and specifically we indicated low (1 or 2 words), middle (3 or 4 words), and high (5 or 6 words) deviators from the position order (Table 2). The majority of participants were within the middle deviation group, with a similar number of participants within the remaining two groups, suggesting a substantial homogeneity in the Time-to-Position task. This pattern was confirmed by the lack of significant differences between these three groups for the mean-adjusted *R*^2^ of the linear fit model (*F*(2,53) = 1.74, *p* = 0.19, *η*^2^*_p_* = 0.06) and the relative b coefficient of the linear model (*F*(2,53) = 1.16, *p* = 0.32, *η*^2^*_p_* = 0.04). Within the regression analysis, we counted how many participants reported a significant negative b coefficient as an index of the bottom-to-top STEARC effect (Table 2). Table 2 shows that there was a similar proportion of participants with and without a significant STEARC effect. Then, we calculated, for each participant, the RT difference between congruent (past words associated with the down arrow and future words associated with the up arrow) and incongruent (past words associated with the up arrow and future words associated with the down arrow) conditions as an index of the direction (e.g., the negative difference indicated a bottom-to-top direction of the mental timeline) and magnitude of the STEARC effect. Thus, we correlated this RT difference and the mean-adjusted *R*^2^ of the linear fit model in the Time-to-Position task. This correlation tended towards significance (*r* = −0.26, *p* = 0.055), suggesting that a larger STEARC effect tended to be associated with a better linear model for explaining the position order found in the Time-to-Position task. Finally, we observed a similar distribution of participants, with low, middle, and high deviations within participants with a significant, non-significant, and reversed STEARC effect (*χ^2^*(4) = 0.82, *p* = 0.94). Indeed, each of the three participants with a reversed STEARC effect was distributed in each of the “deviation” categories. The majority (53.80%) of participants with a significant STEARC effect were middle deviators, while the remaining participants were equally divided into low deviators (23.10%) and high deviators (23.10%). A similar pattern was observed for no-STEARC-effect participants, with 51.90% as middle deviators, 18.50% as low deviators, and 29.60% as high deviators. For further analysis of the relationship between both tasks, see the Appendix A.

### 2.3. Discussion

In Experiment 1, the main result was a vertical STEARC effect, with associations between past and bottom space and between future and top space, confirming previous studies [30,31,32,39,41]. This bottom-to-top orientation of mental time representation was found in both tasks (speeded binary and position tasks). In addition, a slight negative association between the STEARC effect (direction and magnitude) in the binary task and the linear fit model in the Time-to-Position task was found, suggesting that a larger STEARC effect was associated with a higher linear fit for the word order reported in a task with a continuous response format. In the Time-to-Position task, we reported an ordinal sequence of temporal words along the vertical line from “*anticamente*” (the more distant past word) to “*in future*” (the more distant future word), confirming previous studies [31] and extending vertically what has been found horizontally [27]. This last finding was relevant because it gave reliability to this task in its paper-and-pencil version [31] and in its online version (the present study). Moreover, these findings suggest that the temporal words were represented by participants in a specific order, suggesting that each word could be commonly used to refer to different temporal periods in the past and in the future. Moreover, this online version ruled out any influence of radial dimension in performing the task because the PC/laptop screen was placed vertically. Finally, we found a substantial homogeneity in the word position patterns for all participants, indicating that this order was adopted. Altogether these findings could be in line with the account of the “more-is-up” metaphor [44] and, more generally, with the daily experience of an increase of time along vertical space, such as the experience of increasing time to reach the attic of a hotel from the hall.

According to the predicted hypotheses, the ANOVA on RTs confirmed an interaction between the spatial positions of the response keys and temporal words (i.e., STEARC effect) without any significant influence of the spatial positions of the stimuli. Although we found a significant regression coefficient for a linear model, we observed similar proportions of participants with a significant STEARC effect and without an STEARC effect, questioning the consistency of the observed effect. However, we found a correlation between the b coefficient, which displayed a linear model defined by the word order of the Time-to-Position task, and the direction/magnitude of the STEARC effect, suggesting a convergence of the results found in both tasks. In line with Grasso et al. [28] (see also [27] for the horizontal space), the spatial–temporal association was stronger (or observable) when it was implemented via a motor response (or arose at the response selection/execution stage), in a similar way to what has been proposed for numbers [53]. Thus, the results of Experiment 1 seem to confirm and extend along the vertical dimension the findings provided by Fabbri et al. [13,14]. Indeed, the authors explored the possible interaction between the STEARC effect and visuospatial processing along the horizontal space and found an STEARC effect only [13,14]. Furthermore, Vallesi et al. [52] found that the spatial representation of time produced a response code that preactivated the motor cortex with faster responses for congruent mapping (left-short and right-long) than for incongruent mapping. Altogether these findings can be explained according to the computational model of the SNARC effect developed by Gevers et al. [53] for numbers. In this framework, three levels were involved: the mental number line (bottom level), the categorization of the numerical magnitude (middle level), and the response level where the manual responses were spatially coded (top level). In a similar way, we could assume that in our temporal categorization task, the mental timeline (bottom level) was activated to judge whether the temporal words referred to past or future (middle level), and then, the down or up spatial codes of response keys were solved (top level) to perform the task.

However, we did not confirm the RT pattern with NEs, probably reflecting a speed–accuracy tradeoff. In addition, the NE pattern revealed that the spatial position of the temporal words partially influenced the processing of stimuli. Indeed, for future words, we found that their presentation in the upper part of the screen determined fewer errors than their presentation in the remaining two spatial positions. Thus, a marginal influence of the spatial position of the stimuli (visuospatial processing) in the spatial–temporal association seemed to be possible. In other words, the bottom-to-top representation of time was shown in terms of speeded responses and then (later and in a weak way) in terms of accuracy. A possible explanation for these results could be a Simon-like effect (i.e., fewer errors when there was correspondence between the spatial position of the response keys and the spatial position of the stimuli along the screen) which masked any visuospatial effect. Indeed, we found that stimuli presented either at the top or at the center determined fewer errors when they were categorized by the press of the up arrow. Thus, the interaction between Stimulus Location and Response Position could mask the influence of the visuospatial processing, given that we observed fewer errors with the up key and stimuli presented at the top and center of the screen compared to the bottom presentation of the stimuli.

To increase the visuospatial processing of temporal stimuli and assess the heterogeneity of the effect found in the previous experiment, in the subsequent Experiment 2, not only did we present stimuli in three different spatial positions on the screen, but also, they were written in a downward (Experiment 2A) or upward (Experiment 2B) orthography. To the best of our knowledge, this manipulation influenced the vertical space–time association [48] and, thus, this second experiment allowed us to deeper test the purpose of the present study.

## 3. Experiment 2A

### 3.1. Materials and Methods

In Experiment 2A (downward manipulation), a new group of 44 Italian students (33 females; mean age = 24.32 years, SD = 3.21 years) was recruited in exchange for course credit. In line with the previous experiment, the sample size was adequate for within-group comparison and to prevent Type II errors.

According to the EHI score [59], 40 participants were right-handed (M = 90.62, SD = 12.07) and 4 were left-handed (M = −72.85, SD = 39.42). Data from 2 participants (2 females) were excluded from the analysis because these participants reported a percentage of errors higher than 25% (M = 53.19%, SD = 0.59%). The remaining sample was composed of 42 participants (31 females; mean age = 24.48 years, SD = 3.17 years; 38 right-handed). All participants had normal or corrected-to-normal vision and were not informed of the purpose of the study. Informed consent was obtained from all participants and the study was approved by the Ethical Committee of the Department of Psychology at the University of Campania Luigi Vanvitelli.

The stimuli and procedure were the same as in Experiment 1. The only exception was that the stimuli were written from top to bottom (downward direction), as displayed in Figure 3A. As before, participants performed the temporal categorization and the Time-to-Position tasks using the E-Prime 3.0 software (Psychology Software Tools, Pittsburgh, PA, USA).

As in the previous experiment, we considered RTs above and below 3 SDs as outliers and discarded them from the data analysis (about 8% of the sum of trials with errors and outliers). The previous repeated measures ANOVA on RTs and NEs was performed. When a significant effect or interaction was found, the Bonferroni post hoc test was run. The alpha level was set at 0.05.

### 3.2. Results

For the temporal categorization task, all descriptive data for RTs and NEs (with their relative SD) are displayed in Table 3.

The ANOVA on RTs showed a significant Temporal Word effect (*F*(1,41) = 11.31, *p* < 0.05, *η*^2^*_p_* = 0.21), with faster responses for the future (M = 1259 ms; SD = 323 ms) than for the past (M = 1300 ms; SD = 339 ms). In addition, a Stimulus Location effect was found (*F*(2,41) = 25.62, *p* < 0.001, *η*^2^*_p_* = 0.38), with faster responses for stimuli presented at the center (M = 1221 ms; SD = 320 ms) rather than those presented at the bottom (M = 1304 ms; SD = 316 ms) or at the top (M = 1314 ms; SD = 329 ms) of the screen, with *p* < 0.001 for both comparisons. No significant comparison between bottom and top space was revealed in the post hoc test. Moreover, a significant Temporal Word × Response Position interaction was shown (*F*(1,41) = 9.56, *p* < 0.05, *η*^2^*_p_* = 0.19), revealing an STEARC effect: faster RTs with the down key for the past words (M = 1238 ms; SD = 320 ms) than with the up key (M = 1363 ms; SD = 359), with *p* < 0.05, and faster RTs with the up key for future words (M = 1211 ms; SD = 305 ms) than with the down key (M = 1306 ms; SD = 342 ms), with *p* < 0.05. No other significant main effects or interactions were revealed (all *F*s < 1.99 with *p*s > 0.15, *η*^2^*_p_* < 0.02).

The same ANOVA on NEs did not show any main effect or interactions (all *F*s < 0.68 with *p*s > 0.41, *η*^2^*_p_* < 0.02), with the only exception being a significant triple interaction (*F*(2,41) = 4.00, *p* < 0.05, *η*^2^*_p_* = 0.09), as displayed in Table 3. Thus, we performed three Temporal Word × Response Position ANOVAs for each stimulus location separately. In all ANOVAs, no main effects and interactions (STEARC effect) were found (bottom: all *F*s < 1.98 with *p*s > 0.17, *η*^2^*_p_* < 0.05; center: all *F*s < 0.34 with *p*s > 0.56, *η*^2^*_p_* < 0.008; top: all *F*s < 1.48 with *p*s > 0.23, *η*^2^*_p_* < 0.03).

As shown in Appendix B, in the Time-to-Position task, a linear fit model explained the subjective positions of the temporal words along the vertical dimension (*R*^2^ = 0.796, SD = 0.11, *t*(43) = 49.05, *p* < 0.001). Thus, we adopted the same procedure adopted in Experiment 1 for regression analysis. The result showed that the b coefficient (b = −17.61; SD = 38.19) differed significantly from zero (*t*(41) = −2.99, *p* < 0.01, *Cohen’s d* = −0.65), confirming a bottom-to-top representation of the temporal words (Appendix A). To test the presence of individual differences in both tasks, we calculated the number of participants with low, middle, and high deviations from the word order reported in Appendix B. As before, we found that many participants were grouped as middle deviators, and a similar percentage of participants were found in the two remaining groups (Table 4). No group differences were found for the mean-adjusted *R*^2^ of linear fit model (*F*(2,39) = 0.91, *p* = 0.41, *η*^2^*_p_* = 0.05) and the relative b coefficient of the linear model (*F*(2,39) = 1.15, *p* = 0.33, *η*^2^*_p_* = 0.06), suggesting homogeneity of the order pattern. Moreover, we found more participants with non-significant STEARC effect according to the word order reported in the Time-to-Position task (Table 4). Also, we did not find a significant correlation between the adjusted *R*^2^ value of the Time-to-Position task and the direction/magnitude of the STEARC effect (*r* = +0.14, *p* = 0.39). Finally, there was no association between the distribution of participants with low, middle, and high deviation from the word order and those with significant, non-significant, and reversed STEARC effects (*χ^2^*(4) = 4.33, *p* = 0.36). In this case, all participants with a reversed STEARC effect were middle deviators. As before, participants with a significant STEARC effect were equally divided into low (26.70%) and high (26.70%) deviation groups, with the remaining 46.70% as middle deviators. In the non-STEARC effect group, the majority (60.90%) were classified as middle deviators, 26.10% as high deviators and 13.00% as low deviators. For further analysis of the relationship between both tasks, see the Appendix A.

### 3.3. Discussion

Although in Experiment 2A the stimuli were arranged in a downward way (from top to bottom) and displayed in different positions on the screen, we found a vertical STEARC effect in both speeded binary and Time-to-Position tasks. Thus, the result pattern not only confirmed a bottom-to-top representation of time, but also the relevance of the spatial information provided by two response keys for the spatial–temporal association [13,14,16,23,27,28,31]. These considerations were further confirmed by the analysis of NEs, despite the triple interaction. In all spatial positions, no STEARC effect was found, suggesting that the visuospatial processing did not influence the spatial–temporal association. However, in this experiment, the direction and the magnitude of the STEARC effect did not correlate with the linear fit model. Despite the substantial similarity between participants in placing temporal words in a specific order, a possible explanation could be related to the fact that in this experiment the word order in the Time-to-Position task was different from that reported in the same task of Experiment 1 (see Appendix B). Specifically, nine temporal words (three in the past and six in the future) were placed in a different position (e.g., “*una volta*” was in fourth position in Experiment 1 and in second position in the present experiment) with respect to the positions found in Experiment 1.

Interestingly, Experiment 2A failed to replicate the findings displayed by Casasanto and Bottini [48], and probably the present study could question the idea that the spatial–temporal association was culturally mediated and influenced by the reading/writing experience, as suggested by the CORE principle [19]. Our data, however, seemed to add further evidence of the CORE principle but in a broader sense—that is, people experienced the verticality from bottom to top in a series of actions or situations during daily life, creating an association between the experience performed in the environment and the time representation (i.e., a daily experience of vertical space). Moreover, our results could be in line with the Conceptual Metaphor Theory or CMT [44], which assumes a link between the abstract concepts and the concrete sensorimotor experiences through linguistic metaphors, such as the more-is-up metaphor [44]. This further assumption was in line with the previous finding reported by Beracci et al. [32], who reported a bottom-to-top direction of the timeline using different temporal information (i.e., short and long durations within the range of 200–600 ms). Thus, the more-is-up metaphor could explain the preference for a bottom-to-top orientation in temporal representation respect to a reversed direction.

Thus, in the following Experiment 2B, we attempted to assess whether the presentation of temporal stimuli in upward fashion (i.e., in line with a bottom-to-top direction of the mental timeline) could influence the spatial–temporal association vertically. In other words, the experimental manipulation of Experiment 2B should allow us to further test the visuospatial processing effect on the spatial–temporal association.

## 4. Experiment 2B

### 4.1. Materials and Methods

In Experiment 2B (upward manipulation), a new group of 45 Italian students (28 females; mean age = 27.56 years, SD = 8.08 years) was recruited in exchange for a course credit. In line with the previous experiment, the sample size was adequate for within-group comparison and to prevent Type II errors.

Based on the EHI scores [59], 39 participants were right-handed (M = 82.05, SD = 21.91) and 6 were left-handed (M = −68.18, SD = 23.78). Using the previous exclusion criteria, 1 participant (a male) was not included in the data analysis due to a high percentage of errors (93.89%), and thus, the data of the remaining 44 participants (28 females; mean age = 27.59 years, SD = 8.17 years; 38 right-handed) were analyzed. All participants had normal or corrected-to-normal vision and were not informed of the purpose of the study. Informed consent was obtained from all individual participants included and the study was approved by the Ethical Committee of the Department of Psychology at the University of Campania Luigi Vanvitelli.

The experimental stimuli and procedure were the same as those used for Experiment 2A, except for temporal stimuli which were upward orthographically (Figure 3B). All RTs with more than 3 SDs above or below the mean were discarded as outliers (about 7% as the sum of trials errors and outliers). The same statistical analyses adopted in Experiment 2A were performed.

### 4.2. Results

For the temporal categorization task, all descriptive data for RTs and NEs (with their relative SD) are displayed in the Table 3.

The Temporal Word × Response Position × Stimulus Location ANOVA on RTs showed a significant Stimulus Location effect (*F*(2,43) = 75.55, *p* < 0.001, *η*^2^*_p_* = 0.64), with faster responses for stimuli presented at the center (M = 1361 ms, SD = 297 ms) than those at the bottom (M = 1511 ms, SD = 316 ms) or at the top (M = 1451 ms, SD = 317 ms; *p* < 0.001 for both comparisons). A significant Temporal Word × Response Position interaction was found (*F*(1,43) = 6.48, *p* < 0.05, *η*^2^*_p_* = 0.13), suggesting an STEARC effect: faster responses with the down key for past words (M = 1382 ms, SD = 319 ms) than with the up key (M = 1524 ms, SD = 300 ms), with *p* < 0.05, and faster responses with the up key for future words (M = 1360 ms, SD = 309 ms) than with the down key (M = 1497 ms, SD = 312 ms), with *p* < 0.05. Moreover, a significant interaction between Temporal Word and Stimulus Location was found (*F*(2,43) = 3.58, *p* < 0.05, *η*^2^*_p_* = 0.08), revealing faster responses for past words located at the center (M = 1393 ms, SD = 292 ms) than past words located at the bottom (M = 1508 ms, SD = 306 ms; with *p* < 0.0001) or at the top (M = 1457 ms, SD = 330 ms; with *p* < 0.001). In addition, the post hoc test showed that participants produced faster RTs for future words located at the center (M = 1329 ms, SD = 301 ms) than future words located at the bottom (M = 1513 ms, SD = 327) or at the top (M = 1444 ms, SD = 304 ms), with *p* < 0.001 for both comparisons. Finally, we observed faster RTs for past words positioned at the top than those located at the bottom (*p* < 0.05) and for future words positioned at the top than those presented at the bottom (*p* < 0.001). The main ANOVA did not reveal any other significant main effects or interactions (*F*s < 3.29 with *p*s > 0.08, *η*^2^*_p_* < 0.07).

The same ANOVA on NEs showed a significant interaction between Temporal Word and Response Position (*F*(1,43) = 8.83, *p* < 0.05, *η*^2^*_p_* = 0.17), indicating an STEARC effect: fewer errors with the down key for past words (M = 4.44, SD = 6.45) than with the up key (M = 6.26, SD = 7.49), with *p* < 0.05, and fewer NEs with the up key for future words (M = 5.25, SD = 6.74) than with the down key (M = 7.45, SD = 9.17), with *p* < 0.05. Mirroring the RT result pattern, a significant Temporal Word × Stimulus Location interaction was found (*F*(2,43) = 5.73, *p* < 0.05, *η*^2^*_p_* = 0.12). The post hoc test revealed fewer NEs for future words located at the top (M = 5.23, SD = 7.24) than at the bottom (M = 7.76, SD = 9.34), with *p* < 0.05. Although the comparison was not significant, we observed fewer errors for past words when stimuli were presented at the bottom (M = 4.96, SD = 6.54) with respect to stimuli presented at the top (M = 5.90, SD = 7.26). No other significant main effects or interactions were observed (*F*s < 3.05 with *p*s > 0.09, *η*^2^*_p_* < 0.07).

As shown in Appendix B, the word order in the Time-to-Position task fitted well with a linear model (*R*^2^ = 0.809, SD = 0.09, *t*(44) = 58.03, *p* < 0.001) in a bottom-to-top direction. The slope coefficient (b = −19.02, SD = 54.73) differed significantly from zero (*t*(43) = −2.30, *p* < 0.05, *Cohen’s d* = −0.44), confirming the vertical representation of time (Appendix A). Table 4 resumes the individual differences in Experiment 2B. In the Time-to-Position task, more than 50% of participants were middle deviators, with similar percentages of participants in the two remaining groups (Table 4). No group differences were shown for the mean-adjusted *R*^2^ of linear fit model (*F*(2,41) = 1.02, *p* = 0.37, *η*^2^*_p_* = 0.05) and the relative b coefficient of the linear model (*F*(2,41) = 0.36, *p* = 0.70, *η*^2^*_p_* = 0.02), suggesting homogeneity of the order pattern. In the regression analysis, we found a similar number of participants with a negative STEARC effect or non-significant STEARC effect. As before, the correlation between the adjusted *R*^2^ value of the Time-to-Position task and the STEARC effect was not significant (*r* = −0.20, *p* = 0.20), as well as the association between the distribution of participants with low, middle, and high deviation from the word order in the line and those with significant, non-significant, and reversed STEARC effect (*χ^2^*(4) = 5.76, *p* = 0.22). In line with the previous results, 50.00% of participants with a reversed STEARC effect were classified as middle deviators, while 37.50% of participants were classified as low deviators, and only one participant was a high deviator. About 53.00% of participants with a significant STEARC effect were classified as middle deviators, while 35.30% of participants were classified as low deviators and only two participants were high deviators. About 53.00% of participants with no STEARC effect were middle deviators, and about 34% of participants were classified as high deviators, and only two participants were low deviators. For further analysis of the relationship between both tasks, see the Appendix A.

Given the complementary experimental conditions of both Experiments 2A and 2B, we performed a comparison between these two experiments through two mixed ANOVAs with Experiment as a between-subjects factor, and Temporal Word, Response Position, and Stimulus Location as within-subjects factors on RTs and NEs, respectively. When RTs were analyzed, we found a significant Experiment effect (*F*(1,84) = 8.45, *p* < 0.05, *η*^2^*_p_* = 0.09), with faster RTs in Experiment 2A (M = 1285 ms, SD = 290 ms) than those in Experiment 2B (M = 1439 ms, SD = 218 ms). In addition, both Temporal Word (*F*(1,84) = 12.83, *p* < 0.005, *η*^2^*_p_* = 0.13) and Stimulus Location (*F*(2,168) = 85.39, *p* < 0.0001, *η*^2^*_p_* = 0.50) effects were observed. For the former main effect, future words (M = 1344 ms, SD = 317 ms) were categorized more quickly than past words (M = 1377 ms, SD = 325 ms). For the latter effect, stimuli placed in the center (M = 1291 ms, SD = 323 ms) were judged more quickly than those presented at the bottom (M = 1407 ms, SD = 316 ms; *p* < 0.001) and at the top (M = 1382 ms, SD = 323 ms; *p* < 0.001). Moreover, the post hoc test showed a significant comparison between the bottom and top positions (*p* < 0.05). Also, this ANOVA showed significant Experiment × Spatial Position (*F*(2,168) = 8.79, *p* < 0.0001, *η*^2^*_p_* = 0.10), Temporal Word × Response Position (*F*(1,84) = 14.29, *p* < 0.0001, *η*^2^*_p_* = 0.15), and Temporal Word × Stimulus Location (*F*(2,168) = 4.84, *p* < 0.05, *η*^2^*_p_* = 0.05) interactions. The first double interaction mirrored the main Stimulus Location effect in both experiments and the general response speed of Experiment 2A. The second double interaction revealed the STEARC effect with faster RTs for the down arrow in the categorization of past words (M = 1310 ms, SD = 320 ms) than the categorization of future words (M = 1402 ms, SD = 327 ms; *p* < 0.001), and faster RTs for the up arrow in categorizing future words (M = 1286 ms, SD = 307 ms) than past words (M = 1444 ms, SD = 330 ms; *p* < 0.001). The last double interaction revealed a space–time congruency effect in which the post hoc test showed faster RTs for future words presented at the top (M = 1366 ms, SD = 311 ms) than for past words presented in the same position (M = 1398 ms, SD = 335 ms; *p* < 0.05). Beyond faster RTs for future words presented in the center (M = 1261 ms, SD = 323 ms) than for past words (M = 1322 ms, SD = 324 ms; *p* < 0.05), no other significant comparisons were found. No other significant main effect or interactions were observed (*F*s < 3.30 with *p*s > 0.07, *η*^2^*_p_* < 0.04). As regards NEs, no main effects or interactions were observed (*F*s < 3.20 with *p*s > 0.06, *η*^2^*_p_* < 0.04), with the exception of Temporal Words × Stimulus Location (*F*(2,168) = 3.58, *p* < 0.05, *η*^2^*_p_* = 0.04) and Experiment × Temporal Word × Stimulus Location (*F*(2,168) = 4.54, *p* < 0.05, *η*^2^*_p_* = 0.05) interactions. The first interaction revealed that past words presented at the bottom (M = 5.50, SD = 7.57) determined fewer errors than future words presented in the same position (M = 7.24, SD = 9.37; *p* < 0.01), while no other significant comparisons were observed. The second interaction showed that past words presented at the bottom determined fewer errors in Experiment 2B (M = 4.96, SD = 6.55) with respect to Experiment 2A (M = 6.03, SD = 8.60; *p* < 0.05). The same NE pattern was found for past words presented in the center of the screen (Experiment 2A: 6.11 ± 7.81 vs. Experiment 2B: 5.19 ± 7.10; *p* < 0.05). For the future words presented at the top, in Experiment 2B (M = 5.23, SD = 7.25) fewer NEs were reported compared to those reported in Experiment 2A (M = 6.75, SD = 9.43; *p* < 0.05).

### 4.3. Discussion

In Experiment 2B, the stimuli were written upward, and two specific result patterns were found. On the one hand, we confirmed in both tasks the STEARC effect with a bottom-to-top mental timeline. Thus, we confirmed the idea that the daily experience of “pilling objects” could explain the way in which “time increases from the ground to the floor”, probably in line with the more-is-up metaphor [44]. However, we found for RTs and NEs a significant interaction between Temporal Word and Stimulus Location, suggesting a visuospatial processing effect on the task performance. Despite a general advantage for the center position, we could speculate that the upward orthography of the temporal words could facilitate the processing of temporal stimuli. Indeed, participants reported fewer NEs for past words located in the bottom space and fewer NEs for future words located in the top space, suggesting a space–time congruency effect. In other words, the correspondence between the upward fashion of stimuli, bottom-to-top position of the stimuli, and bottom-to-top direction of the mental timeline could influence the processing of temporal information. This speculative interpretation could add evidence for a bottom-to-top mental representation of time. This assumption could be advanced because only in this Experiment 2B we observed a specific influence of the visuospatial processing on spatial–temporal association, probably due to the upward direction of temporal stimuli. When we compared the data of Experiment 2B to those of Experiment 2A, we confirmed the idea that an upward orthography influences a visuospatial processing effect, especially for accuracy. However, in Experiment 2B, we confirmed the lack of correlation between the direction/magnitude of the STEARC effect and the linear fit model of the Time-to-Position task. Although we found a similar distribution of participants in low, middle, and high deviation groups, Appendix B reveals that in this experiment the word order differed from that of Experiment 1 in six words (three for the past and three for the future), suggesting that any deviation from the “real” word order of Experiment 1 could limit any relationship between speeded binary and continuous position tasks.

It is worth noting that in this study we also failed to replicate the findings provided by Casasanto and Bottini [48], questioning the hypothesis that writing and reading habits (or exposure) are relevant for the spatial–temporal mapping. A possible explanation for this failure could be related to the lack of any training with downward and upward orthography during task instruction [48] (see also [19]). Future studies should investigate this point deeply. Nevertheless, participants of Experiment 2A were quicker to respond to temporal stimuli than those of Experiment 2B, suggesting a type of reading-writing effect on the processing of stimuli. This top-to-bottom habit could interfere with the bottom-to-top direction of the timeline.

## 5. General Discussion

The purpose of the present study was to test possible interactions between the STEARC effect and visuospatial processing. In addition, all experiments were designed to respond to inconclusive data found in the literature on the orientation of the mental timeline vertically. In all studies, a clear STEARC effect was found, and this effect was further confirmed in the joint analysis performed to compare Experiments 2A and 2B. By contrast, a visuospatial processing effect was not found systematically with specific cases in an accurate pattern. In the subsequent part of this section, we discuss our results separately.

### 5.1. Vertical STEARC Effect

In all experiments, the spatial–temporal association was mainly influenced by the spatial position of the response keys, with associations between past words and the down key, as well as between future words and the up key. This STEARC effect could be explained by the polarity correspondence account proposed by Proctor and Cho [50]. In short, the polarity correspondence account suggests that short durations or past events are associated with negative polarities, whereas long durations or future events are associated with positive polarities. These polarities are also assigned to response alternatives (e.g., left or right key or down or up key), and thus the matching between the polarity of the stimulus and the polarity of the response key determines faster RTs than unmatching conditions. Thus, the STEARC effect found here in all experiments resembled the correspondence between past words (negative polarity) and the down key (negative polarity) as well as between future words (positive polarity) and the up key (positive polarity). However, Santiago and Lakens [62] failed to support the polarity correspondence account as a possible framework for explaining the spatial–temporal association. Moreover, the similarity of the findings between the present research vertically and those reported by Grasso et al. [28] horizontally could rule out the polarity correspondence account for our data.

An alternative account suggested that the vertical STEARC effect found here could be more related to the response selection/execution stage than to the visuospatial processing stage [14,23,27,28,31,32,52,53] when manual responses and physical codes were involved at the same time. In line with the computational model of the SNARC effect [53], the response selection/execution stage solved the associations between past-future stimuli and down-up keys. Therefore, our data reinforced the hypothesis that the vertical spatial–temporal association seemed to be mainly related to the response mapping rather than the stimuli location. This assumption could be in line with neuroimaging data with an involvement of the motor cortex for the STEARC effect [51,52]. Thus, our data partially confirmed the ATOM model [3,4,5] due to the involvement of the prefrontal cortex in temporal processing [3,4,5]. Although we did not directly test the ATOM model at the neural level, we did not find a space–time congruency effect in all experiments and, thus, the involvement of PPC and IPS can be questioned. A possible interpretation of the lack of a space–time congruency effect could be related to the type of temporal information used in the present study (see Núñez and Cooperrider [63] for a definition of temporal stimuli used in the literature). Indeed, the ATOM model is mainly based on the definition of time as quantity, such as temporal durations [13]. The use of past- and future-related words which did not involve any quantity information could have limited the possibility of the involvement of PPC and IPS in creating the space–time congruency effect [51].

### 5.2. Vertical Visuospatial Processing Effect

This type of effect was only found when NEs were analyzed, although this result pattern was not systematic through the experiments. Indeed, in Experiment 1, past words were associated with bottom space, whereas future words were associated with top space. In Experiment 2A, there was a triple interaction, but the STEARC effect was not modulated by spatial information. In Experiment 2B, the NE pattern mirrored the results found in Experiment 1. When we compared Experiments 2a and 2b, we confirmed a general association between past-bottom and future-top, particularly in Experiment 2B (i.e., upward presentation of the stimuli). Thus, the (unconventional) presentation of temporal words from the bottom to top could facilitate the space–time congruency effect, given that there is a “congruency” between the direction of the mental timeline and the spatial–temporal associations in physical space. On the one hand, these results of the NEs could indicate that the spatial information could decay [64] or be actively suppressed [65] in the time course of the formation of the time-related spatial information (see also Gevers et al. [66] for a similar explanation of the spatial–numerical association). On the other hand, our results of the NEs could be in line with the intermediate coding account [49]. In line with Fabbri et al. [14,23], the spatial–temporal association was automatically activated by the spatial positions of the response keys, probably due to a preexisting association in the long-term memory, and then later, this spatial–temporal association was influenced by the spatial position of the stimuli, given that the spatial codes of both manual responses and position sides of the screen were involved in the task at the same time. Future studies should address these assumptions using, for example, durations as Fabbri et al. did [14,23].

### 5.3. Vertical Mental Timeline

All experiments in both tasks showed a bottom-to-top time representation in line with other studies in temporal [30,31,32,38,39,41] and numerical [1,53,54,67,68,69,70] domains. For both time and numbers, this vertical direction could be explained by people’s experiences in daily life of the more-is-up metaphor [44,45,46]. This assumption is also reinforced by the Grounded Theory [71] and TEST model [45], suggesting that abstract concepts along the vertical space are based on the concrete sensorimotor experience of the physical world. Thus, specific experiences, such as to fill a glass of water or to take the lift from the bottom to the top floor, could reinforce a vertical bottom-to-top representation of magnitudes. However, Dalmaso et al. [72] recently showed that the direction of the vertical STEARC effect could depend on how time-related stimuli are distributed, probably affecting how they are represented. According to the authors, time durations (i.e., shorter or longer than a reference stimulus) should probably activate a bottom-to-top time representation, whereas time defined as a short–long dichotomy could activate a top-to-bottom representation. Thus, future studies should confirm the present study with different temporal stimuli and task paradigms. Moreover, future studies should address the involvement of reading and/or writing habits, as well as our routine of scrolling downward through mailboxes or text messages, in spatial–temporal associations, given that we observed in the joint analysis faster RTs for categorizing temporal words when they were presented downward in line with the direction of our reading habit.

### 5.4. The Time-to-Position Task and Its Association with the Temporal Categorization Task

A particular mention is reserved for the data of the Time-to-Position task: all past words were below, and all future words were above the middle of the line; we also found that participants placed temporal words in a specific order vertically. These findings could suggest that the vertical spatial–temporal associations could be discussed within the framework of the position effect (e.g., refs. [73,74,75]). In line with van Dijck and Fias [73] for spatial–numerical association, it is possible to speculate that the spatial position of each temporal word is represented and maintained in the working memory [74]. In other words, the word order could derive from the involvement of the working memory in associating temporal words with space, especially using vertical coordinates in line with the experience of verticality [73,74,75]. This assumption could be based on the evidence that in all the experiments, a significant b coefficient was found when we regressed the RT difference between up and down arrows for each word according to the word order found in the corresponding Time-to-Position task. Interestingly, several studies have shown for the numerical domain a double dissociation between the type of working memory load (verbal or visuospatial) and the type of task (explicit or implicit), showing that in explicit tasks, the spatial–numerical association disappeared with a visuospatial load while during an implicit task the SNARC effect disappeared with a verbal load. Future studies should investigate this possible dissociation for the temporal domain (see Ginsburg et al. [75] for a review of this topic).

However, the performance in the Time-to-Position task was weakly associated with what has been observed in the speeded binary task. Although we reported homogeneity across experiments for the Time-to-Position task (see the boxplots in the Appendix B) as well as similarity in the individual differences for the word order provided, only in Experiment 1 did the correlation between the direction/magnitude of the STEARC effect and the adjusted *R*^2^ of the linear fit model tend towards significance. In all experiments, about 50% of participants reported a non-significant b coefficient, probably explaining the differences across the studies in the word order (e.g., in Experiment 2A, nine words differed in the position from those placed into the line in Experiment 1). Another aspect was related to the findings regarding a more categorical than continuous fashion of describing the data pattern (see Appendix A). As reported by Beracci et al. [27] for the horizontal dimension, the mental timeline is continuous and can be used in a categorical way when explicit temporal information (e.g., a categorization task or Time-to-Position task) is required. This assumption is further based on the evidence provided by Gevers et al. [53] who have demonstrated that a magnitude comparison task induces a categorical representation of numbers, whereas a parity judgment task induces a continuous representation of a mental number line. Thus, the continuous representation of the mental timeline could be used categorically according to the type of task, such as the categorization of temporal words as past or future in binary decision-making. Future studies should address this point in a deeper way, probably using other definitions of time [63] and other type of tasks.

### 5.5. Limits

The present research was not exempt from limits. First, all three experiments were performed online during the pandemic lockdown in Italy, reducing the experimental control, which is usually achieved in a laboratory setting. For example, our online procedure did not ensure that participants followed task instructions (e.g., fingers used to press the keys). In addition, in a laboratory setting the use of more vertically oriented response keys without semantic references to the vertical direction should be adopted, although Beracci and Fabbri [31] have already demonstrated the validity of using arrow keys for the vertical dimension (excluding any alternative explanations, such as a radial axis). Finally, the COVID-19 lockdown limited the possibility to test changes in the strength and direction of the STEARC effect for tasks involving vertical spatial processing due to the presence of landmarks or environmental cues (e.g., the up and down arrows of the lift or the illustrative signs of where the departments are located on the various floors of a hospital, such as the cardiology department on the fourth floor). Future studies should replicate our experiments in a laboratory setting with greater experimental control of the procedure and/or response keys displayed physically with a vertical orientation. Moreover, several experiments in which participants are firstly subjected to various experiences of vertical visuospatial processing (e.g., Tom Cruise climbing the mountain wall in *Mission Impossible 2*) and then are requested to judge temporal stimuli, are recommended to address how visual context influences the STEARC effect.

Second, the list of stimuli encompassed words which vary in morphology and length, ranging from short single to long words. Although these temporal stimuli have been successfully used and controlled for different variables [27,31], and although the program used to design the experiments allowed us to standardize the stimuli size as a single dimension in different screens, future studies should define the temporal stimuli in order to make the comparisons between and among studies more accurate. Third, future studies should use our Time-to-Position task without fixed flankers, which might limit a genuine association and introduce a bias into the task. Alternatively, future studies should propose a vertical line with the opposite flankers (i.e., distant future at the bottom and distant past at the top) in order to strengthen our findings. Fourth, individual differences should be taken into account, considering, for example, how the daily use of mobile phones, emails, or social media may influence our spatial representation of time due to habitual behaviors of scrolling downward or upward to check new or old messages. In line with this possible limitation, a control task for habitual scrolling activity should be provided in future studies in order to rule out the impact of this habitual motor activity on the representation of time along the vertical axis. For example, it is possible to design an experiment in which participants are requested to press up and down arrow keys in order to categorize non-temporal information, such as type of font or capital letters. In this case, it would be possible to control for the impact of the habitual direction of scrolling down. Finally, it should be acknowledged that all studies were performed in Italy and thus, no possible cultural differences in the strength and direction of the STEARC effect during visuospatial processing along the vertical space could be tested. Considering that it has been repeatedly found that in Asiatic culture with a Mandarin language, the vertical STEARC effect is similar to (or even stronger than) the horizontal STEARC effect due to the habitual experience of writing and reading vertically [12,18,20,21,33,34,35,37,38,47,48], a replication of the present study in different cultures is suggested.

## 6. Conclusions

In this study, participants performed an online temporal task, judging 20 Italian temporal words referring to the past and future located at the bottom, center, and top side of the screen, and written in a canonical, upward, and downward way, using down and up arrows on a standard keyboard. In addition, participants performed a Time-to-Position task in which all temporal words were placed along a vertical line, using a continuous response format. The results of three experiments showed a clear vertical STEARC effect, with an association between the past and the down arrow, as well as the future and the up arrow, suggesting a bottom-to-top vertical mental timeline. These findings were further confirmed by the temporal words ordered along a vertical line. The influence of visuospatial processing on spatial–temporal association seemed to be weak (and not constant), and, probably, there is a stronger influence from the response selection/execution stage (i.e., the manual response selection) than the visuospatial processing stage (i.e., the position of the stimuli on the screen) on the spatial–temporal association. This bottom-to-top timeline could be related to our daily life experiences in the physical world (e.g., a plant grows upward), more in line with the metaphor of more-is-up. A possible application of the present study could be found in the organization of a website, placing recent news and future events at the top of the site, and older news and past events at the bottom of the website. We think that our data could be applied in several fields of human–computer interaction or virtual reality.

## Figures and Tables

**Figure 1 brainsci-14-00184-f001:**
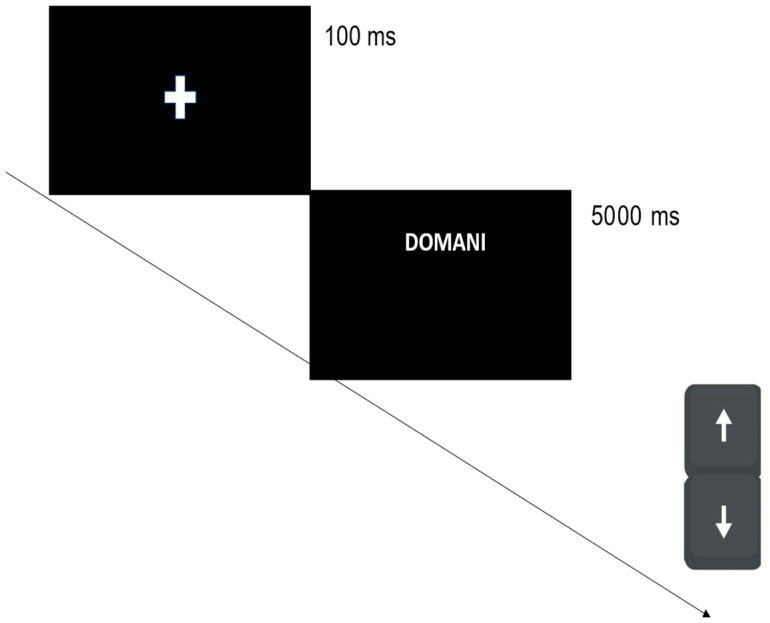
The sequence of the trial in the temporal categorization task adopted in Experiment 1. Participants indicated the temporal reference of each target presented in different spatial positions (top, center, bottom side of the screen), pressing the down-up arrows of a standard keyboard.

**Figure 2 brainsci-14-00184-f002:**
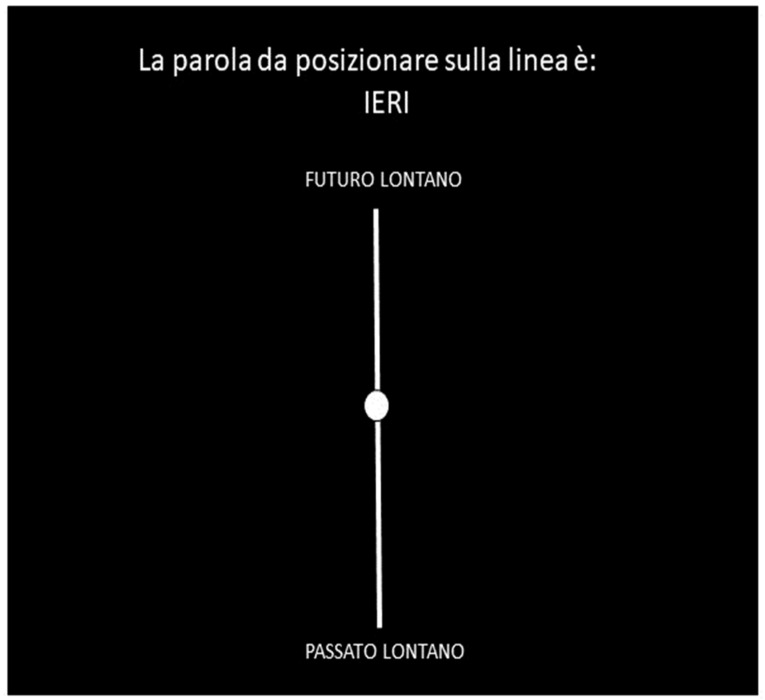
Example of the online version of the Time-to-Position task. When the target word to be placed into the line was presented (e.g., IERI), participants had to move the central cursor (e.g., the white circle at the center of the line) with the mouse downward or upward to indicate the position of the target word along the line.

**Figure 3 brainsci-14-00184-f003:**
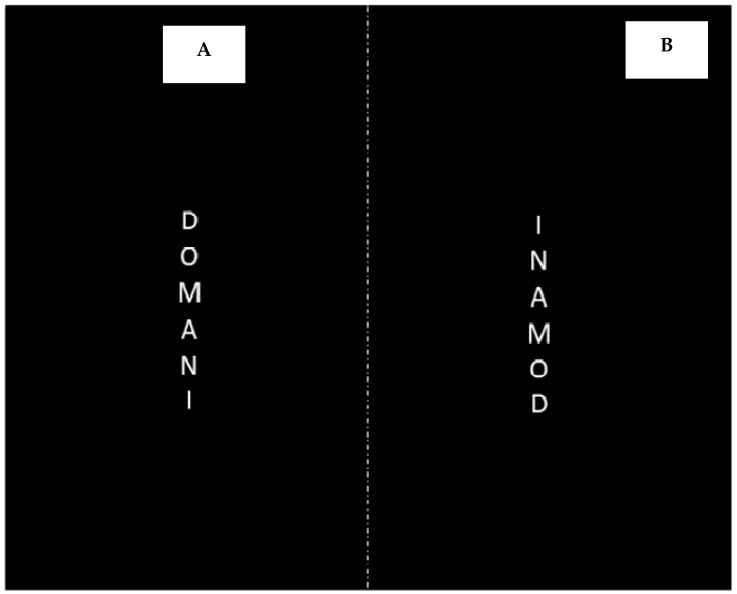
(**A**) An example of the presentation of the stimuli in Experiment 2A; (**B**) an example of the presentation of the stimuli in Experiment 2B.

**Table 1 brainsci-14-00184-t001:** The mean RTs and NEs (and their SDs) for past and future words presented at the bottom, center, and top of the screen, judged by pressing the down and up arrow keys in Experiment 1.

RTs	Down Key			Up Key		
	Bottom	Center	Top	Bottom	Center	Top
**Past words**	1097	1031	1116	1208	1170	1208
	(281)	(263)	(262)	(275)	(308)	(297)
**Future words**	1222	1171	1240	1118	1001	1090
	(310)	(294)	(301)	(261)	(264)	(282)
**NEs**	**Down Key**			**Up Key**		
	**Bottom**	**Center**	**Top**	**Bottom**	**Center**	**Top**
**Past words**	3.15	4.46	5.48	6.07	5.65	5.48
	(3.51)	(4.85)	(5.33)	(5.86)	(6.03)	(5.41)
**Future words**	7.02	6.19	6.49	7.80	4.11	3.87
	(8.38)	(6.18)	(6.89)	(9.62)	(4.49)	(4.39)

**Table 2 brainsci-14-00184-t002:** The first three columns indicate the number of participants with low, middle, and high word deviations from the word order of the Time-to-Position task, and their relative mean-adjusted *R*^2^ and b coefficients are reported. The subsequent columns indicate the number of participants with a significant STEARC effect (negative b coefficient), a non-significant effect (non-significant b coefficient), and a significant reversed STEARC effect (positive b coefficient) reported. The mean-adjusted *R*^2^ of the regression linear model defined by the word order of the Time-to-Position task, the mean b value, and the mean RT difference (i.e., the direction and the magnitude of the STEARC effect) between congruent and incongruent conditions are also reported. Within the parentheses, the SD is reported.

Time-to-Position Tsk	*N*	Mean-Adjusted *R*^2^ Fit Model	Mean b Coefficient Fit Model	Regression Analysis	*N*	Mean-Adjusted *R*^2^	Mean b Coefficient	Direction and Magnitude of Effect
**Low deviators**	12	0.83	+3.84	STEARC effect	26	0.35	−42.99	−583 ms
		(0.08)	(0.60)			(0.12)	(20.11)	(300 ms)
**Middle deviators**	29	0.81	+4.16	No STEARC effect	27	0.03	−2.09	−5.00 ms
		(0.12)	(0.79)			(0.06)	(15.79)	(215 ms)
**High deviators**	15	0.74	+4.31	Reversed STEARC effect	3	0.20	+28.09	+332 ms
		(0.19)	(0.95)			(0.06)	(14.17)	(160 ms)

**Table 3 brainsci-14-00184-t003:** The mean RTs and NEs (and their SDs) for past and future words presented at the bottom, center, and top of the screen, judged by pressing the down and up arrow keys in Experiment 2.

	Experiment 2A						Experiment 2B					
RTs	Down Key			Up Key			Down Key			Up Key		
	Bottom	Center	Top	Bottom	Center	Top	Bottom	Center	Top	Bottom	Center	Top
**Past words**	1233	1189	1291	1392	1309	1388	1441	1330	1375	1575	1457	1540
	(310)	(342)	(307)	(335)	(370)	(373)	(322)	(294)	(342)	(290)	(291)	(318)
**Future words**	1341	1246	1332	1250	1141	1243	1580	1391	1521	1445	1267	1367
	(338)	(356)	(331)	(280)	(330)	(304)	(334)	(285)	(318)	(320)	(318)	(290)
**Experiment 2A**							**Experiment 2B**					
**NEs**	**Down Key**			**Up Key**			**Down Key**			**Up Key**		
	**Bottom**	**Center**	**Top**	**Bottom**	**Center**	**Top**	**Bottom**	**Center**	**Top**	**Bottom**	**Center**	**Top**
**Past words**	7.14	6.63	5.62	4.94	5.54	5.96	4.47	3.86	5.00	5.45	6.51	6.82
	(12.02)	(11.04)	(12.50)	(5.21)	(4.54)	(6.97)	(6.74)	(5.84)	(6.76)	(6.35)	(8.35)	(7.77)
**Future words**	5.22	5.86	6.80	8.15	6.44	6.03	9.39	6.97	5.98	6.14	5.15	4.47
	(5.74)	(6.43)	(6.97)	(13.03)	(11.06)	(11.82)	(10.74)	(8.16)	(8.61)	(7.94)	(6.41)	(5.88)

**Table 4 brainsci-14-00184-t004:** Separately for Experiments 2A and 2B, the first three columns indicate the number of participants with low, middle and high word deviations from the word order of the Time-to-Position task, as well as their relative mean-adjusted *R*^2^ and b coefficients. The subsequent columns indicate the number of participants with a significant STEARC effect (negative b coefficient), with non-significant effect (non-significant b coefficient), and with a significant reversed STEARC effect (positive b coefficient). The mean-adjusted *R*^2^ of the regression linear model defined by the word order of the Time-to-Position task, the mean b value, and the mean RT difference (i.e., the direction and the magnitude of the STEARC effect) between congruent and incongruent conditions are also reported. Within the parentheses, the SD is reported.

Experiment 2A								
Time-to-Position Task	*N*	Mean-Adjusted *R*^2^ Fit Model	Mean b Coefficient Fit Model	Regression Analysis	*N*	Mean-adjusted *R*^2^	Mean b Coefficient	Direction and Magnitude of Effect
**Low deviators**	7	0.85	+3.73	STEARC effect	15	0.33	−53.30	−678 ms
		(0.05)	(0.41)			(0.14)	(38.65)	(407 ms)
**Middle deviators**	25	0.79	+4.11	No STEARC effect	23	0.0001	−2.71	−24 ms
		(0.13)	(0.68)			(0.05)	(14.91)	(197 ms)
**High deviators**	10	0.79	+4.19	Reversed STEARC effect	4	0.34	+30.62	+369 ms
		(0.10)	(0.67)			(0.11)	(12.87)	(219 ms)
**Experiment 2B**								
**Time-to-Position task**	*N*	Mean-adjusted *R*^2^ Fit model	Mean b coefficient fit model	Regression Analysis	*N*	Mean-adjusted *R*^2^	Mean b coefficient	Direction and Magnitude of effect
**Low deviators**	11	0.80	+3.94	STEARC effect	17	0.43	−72.42	−973 ms
		(0.10)	(0.88)			(0.12)	(24.81)	(371 ms)
**Middle deviators**	23	0.81	+4.02	No STEARC effect	19	0.02	−7.02	−128 ms
		(0.09)	(0.60)			(0.06)	(14.07)	(261 ms)
**High deviators**	10	0.76	+4.19	Reversed STEARC effect	8	0.46	66.10	+831 ms
		(0.11)	(0.71)			(0.17)	(29.79)	(373 ms)

## Data Availability

This study was not preregistered, and the data of this work can be found on OSF at the following link: https://osf.io/yg7pv/?view_only=2910411faeb3420181e32af8bc0f60e1.

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
