# Peer review of "Vertical Mental Timeline Is Not Influenced by VisuoSpatial Processing"

_brainsci, 2024, doi:10.3390/brainsci14020184_

Round 1
Reviewer 1 Report (New Reviewer)
Comments and Suggestions for Authors
This study investigated whether and how visuospatial processing stage modulated the vertical timeline in an online temporal categorization task.
This study is interesting and well-written.
Please add also to you study the organisation name, date and protocol number of the permission of all your experiments.
Author Response
Reviewer#1:
-“This study investigated whether and how visuospatial processing stage modulated the vertical timeline in an online temporal categorization task. The study is interesting and well-written. Please add also to you study the organisation name, date and protocol number of the permission of all your experiments”.
We thank the reviewer#1 for their positive comments. According to the template word file of the Journal, all reviewer’s requests are placed at page 22 as follows: “Institutional Review Board Statement: “The study was conducted in accordance with the Declaration of Helsinki and approved by the Ethics Committee of Department of Psychology, University of Campania Luigi Vanvitelli (protocol code 10/2019 approved March 26, 2019).” In this revised version of the paper, this part has been highlighted in red in order to help the reviewer#1 to detect it.
Reviewer 2 Report (New Reviewer)
Comments and Suggestions for Authors
Review on the manuscript of Beracci A & Fabbri M: “Vertical Mental Timeline Is Not Influenced by VisuoSpatial Processing”.
In this study, Authors explored the possible interaction between Spatial-TEmporal Association of Response Codes (STEARC) effect and visuospatial processing along the vertical space.
Overall, I felt that this topic is of great interest. The results are clearly presented, and the manuscript is well written, in general. In addition, I consider that the manuscript is precise on the question that Authors proposed to study. Thus, the issues that arise to me are listed below, so, I hope Authors find the following comments and suggestions useful.
1 – The manuscript is very extensive, so it would be very easy to the readers lose the focus. It would be better if Authors could simplify the writing in some sections.
2 - An intriguing question is: are there cultural differences in the strength and direction of the STEARC effect during visuospatial processing along the vertical space? Authors tested groups of individuals from the same country. Can Authors elaborate on this in the discussion section and explain if they consider this point a limitation of the study?
3 - Another question is how STEARC effect applies in practical domains, such as human-computer interaction or virtual reality. Can Authors discuss this idea?
4 – Another important consideration is to what extent the visual context, such as the presence of landmarks or environmental cues, influence the strength and direction of the STEARC effect in tasks involving vertical spatial processing. The study was performed during the COVID-19 lockdown, so in a very particular period of abnormal social interaction. It would be good if Authors could discuss this idea, as future research topics to explore.
Comments on the Quality of English LanguageSome minor mistakes detected.
Author Response
Reviewer#2:
-“Review on the manuscript of Beracci A & Fabbri M: “Vertical Mental Timeline Is Not Influenced by VisuoSpatial Processing”. In this study, Authors explored the possible interaction between Spatial-TEmporal Association of Response Codes (STEARC) effect and visuospatial processing along the vertical space. Overall, I felt this topic is of great interest. The results are clearly presented, and the manuscript is well written in general. In addition, I consider that the manuscript is precise on the question that Authors proposed to study. Thus, the issue that arise to me are listed below, so, I hope Authors find the following comments and suggestions useful”.
We would like to thank the reviewer#2 for their positive comments about the manuscript in general. Also, we would like to thank the reviewer#2 for their comments and suggestions we found useful and all changes in the text have been highlighted in red.
-“1. The manuscript is very extensive, so it would be very easy to the readers lose the focus. It would be better if Authors could simply the writing in some sections”.
We tried to simplify the writing through the manuscript. All changes are in red. Minor mistakes are corrected.
-“2. An intriguing question is: are there cultural differences in the strength and direction of the STEARC effect during visuospatial processing along the vertical space? Authors tested groups of individuals from the same country. Can Authors elaborate on this in the discussion section and explain if they consider this point a limitation of the study?”
In the General Discussion at page in the paragraph 5.5 we elaborate possible cultural differences for the strength and the direction of the STEARC effect during visuospatial processing, highlighting this as a possible limit of the present study.
-“3. Another question is how STEARC effect applies in practical domains, such as human-computer interaction or virtual reality. Can Authors discuss this idea?”
In the Conclusion paragraph (the last sentence), we tried to advance a possible application of our data for human-computer interaction.
-“4. Another important consideration is to what extent the visual context, such as the presence of landmarks or environmental cues, influence the strength and direction of the STEARC effect in task involving vertical spatial processing. The study was performed during the COVID-19 lockdown, so in a very particular period of abnormal social interaction. It would be good if Authors could discuss this idea, as future research topics to explore”.
In the limit paragraph (5.5) of the General Discussion, we highlighted this point as a possible limit, and we tried to give insightful inputs for future research.
Reviewer 3 Report (New Reviewer)
Comments and Suggestions for Authors
The authors investigated whether a vertical timeline and its cognitive processing were related to visuo-spacial cues or to other information used for cognitive processing. Results showed that words indicating a past event were cognitively elaborated faster, when the adjacent arrow represented ‘down’; accordingly, words indicating future events were cognitively elaborated faster, when the adjacent arrow represented ‘up’. Such patterns of cognitive elaboration were independent from the positions of the specific words.
The study idea is smart.
Abstract: While the authors mentioned three studies, the text does not help to identify which parts of the results belonged to which study. Please clarify.
Shouldn’t the wording be: ‘bottom-up’ and ‘top-down’? - at least, it’s the first time to read ‘bottom-to-top’ and ‘top-to-bottom’; please clarify.
“In literature, the results, about a mental timeline along a vertical axis, are not conclusive, especially regarding its direction (bottom-to-top vs. top-to-bottom).” Please revise this sentence for grammar and stile.
The last sentence of the Abstract was particularly helpful! Congrats on you!
Introduction:
“…. with a time-related words..” please correct.
Overall, the Introduction section provides a thoroughly crafted overview of the topic, the state-of-the-art and above all the overview of possible alternative explanations. Congrats on you!
Method section.
The three studies were consecutively presented and explained in a thoroughly crafted fashion.
Discussion: As the Introduction, the Discussion section is very well crafted and offers a broad variety of alternative interpretations of the results. Congrats on you!
Comments on the Quality of English Languagenihil
Author Response
Reviewer#3:
-“The authors investigated whether a vertical timeline and its cognitive processing were related to visuospatial cues or to other information used for cognitive processing. Results showed that words indicating a past event were cognitively elaborated faster, when the adjacent arrow represented ‘down’; accordingly, words indicating future events were cognitively elaborated faster, when the adjacent arrow represented ‘up’. Such patterns of cognitive elaboration were independent from the position of the specific words. The study idea is smart”.
We would like to thank the reviewer#3 for correctly summarizing the paper and for the positive comment about it.
-“Abstract: While the authors mentioned three studies, the text does not help to identify which parts of the results belonged to which study. Please clarify”.
We have clarified it in the Abstract (the changes are in red).
-“Shouldn’t the wording be: ‘bottom-up’ and ‘top-down’? – at least, it’s the first time to read ‘bottom-to-top’ and ‘top-to-bottom’; please clarify”.
In reality, the expressions bottom-to-top and top-to-bottom are commonly used in this specific literature topic, probably to disentangle them from bottom-up and top-down types of perceptual processing or types of cognitive information-processing. In the present study (as well as in the literature) the bottom-to-top and top-to-bottom indicate the direction of time representation and cognitive processing along a specific orientation.
-“In literature, the results, about a mental timeline along a vertical axis, are not conclusive, especially regarding its direction (bottom-to-top vs. top-to-bottom)”. Please revise this sentence for grammar and stile.
Done.
-“The last sentence of the Abstract was particularly helpful! Congrats on you!”
Thank you very much for positive comment.
-“Introduction: “…with a time-related words…” please correct”.
Done.
-“Overall, the Introduction section provides a thoroughly crafted overview of the topic, the state-of-the-art and above all the overview of possible alternative explanations. Congrats on you!”
Thank you very much for this positive comment.
-“Method section. The three studies were consecutively presented and explained in a thoroughly crafted fashion”.
Thank you very much for this positive comment.
-“Discussion: As the Introduction, the Discussion section is very well crafted and offers a broad variety of alternative interpretations of the results. Congrats on you!”
Thank you very much for this positive comment.
Round 2
Reviewer 1 Report (New Reviewer)
Comments and Suggestions for Authors
Thank you for all corrections.
This manuscript is a resubmission of an earlier submission. The following is a list of the peer review reports and author responses from that submission.
Round 1
Reviewer 1 Report
Comments and Suggestions for Authors
Dear Editor,
I apologise to you and the authors for this slow response to the revision of Brainsci 2753791… I have been sick.
.
The authors have made a major effort to answer my critique (I am ref 1) and the paper is now in much better shape, being easier to follow.
Two issues remain. The first is that the authors removed Fig 7 and only provide a regression analysis. Without Fig. 7, however, it is impossible for the reader to tell whether a progressive changes, as implied by linear regression, or a binary split, is a better description of the data.
Just looking at Fg 7 in the first draft, it was obvious to me (and to ref 3) that a binary split is the better description. To show this, I suggested calculating a root mean square (RMS) indication of fit. For example, calculate the square-root of the sum of the squared deviations between (a) all the data and the best-fit straight line (i.e., the regression), for comparison with (b) the square-root of the sum of the squared deviations between all the data below the split and its mean value plus all the squared deviations between the data above the split and its mean value, making sure to use the same code so as to have truly equivalent values. I will be shocked if the RMS for linear regression is smaller than that for the binary split. (Since each analysis has two free parameters (mean and slope for regression, two means for the binary split), and applies to the same number of data points, there is no need to correct for the degrees of freedom, which will be equal.) At all events, there is no reason to remove the graph; it should stay, ideally with both types of predictions (regression and binary split) shown with the data.
The more basic issue is, however, that the authors have apparently not run a control for the typical direction of scrolling, so they have not shown that the results have anything to do with a mental time line rather than a motor habit. I had suggested running the experiment on old people, but most likely this will not work as everyone has been scrolling nowadays, old and young. So how about using non-temporal information, such as type of font, and testing the differences between reaction times for upwards and downwards scrolls ? The authors need to think about a suitable control for the motor habit, before I would be positive for publication. I hope that they do, because otherwise this is a fine piece of work and could be a real contribution to the field.
Author Response
Reviewer#1:
-“The authors have made a major effort to answer my critique (I am ref 1) and the paper is now in much better shape, being easier to follow”.
We would like to thank the reviewer#1 for her/his positive comment highlighting the effort we put on the manuscript in the first round of the review.
-“Two issues remain. The first is that the authors removed Fig 7 and only provide a regression analysis. Without Fig. 7, however, it is impossible for the reader to tell whether a progressive changes, as implied by linear regression, or a binary split, is a better description of the data. Just looking at Fig 7 in the first draft, it was obvious to me (and to ref 3) that a binary split is the better description. To show this, I suggested calculating a root mean square (RMS) indication of fit.For example, calculate the square-root of the sum of the squared deviations between (a) all the data and the best-fit straight line (i.e., the regression), for comparison with (b) the square-root of the sum of the squared deviations between all the data below the split and its mean value plus all the squared deviations between the data above the split and its mean value, making sure to use the same code so as to have truly equivalent values. I will be shocked if the RMS for linear regression is smaller than that for the binary split. (Since each analysis has two free parameters (mean and slope for regression, two means for the binary split), and applies to the same number of data points, there is no need to correct for the degrees of freedom, which will be equal.) At all events, there is no reason to remove the graph; it should stay, ideally with both types of predictions (regression and binary split) shown with the data”.
We completely agree with the idea that the data shown by regression analysis are better described by a binary function while the linear model is not the best way to describe them. As stated in the first round, we think that this information is not a goal of the paper, but an attempt to unify both tasks used in the manuscript. Thus, we decided to insert as Supplementary Materials the figures of the regressions with the RMSE for linear and binary split predictions (and the suggested analysis of the reviewer#3 in the first round). However, in the past revised version (see reviewer #3 now #2 in this second round), we already discussed the idea of a categorical (binary) description of the regression, limiting any assumptions about the linearity (see pages 20-21, lines 891-908).
“The more basic issue is, however, that the authors have apparently not run a control for the typical direction of scrolling, so they have not shown that the results have anything to do with a mental time line rather than a motor habit. I had suggested running the experiment on old people, but most likely this will not work as everyone has been scrolling nowadays, old and young. So how about using non-temporal information, such as type of font, and testing the differences between reaction times for upwards and downwards scrolls ? The authors need to think about a suitable control for the motor habit, before I would be positive for publication. I hope that they do, because otherwise this is a fine piece of work and could be a real contribution to the field”.
In this case, we disagree with the reviewer#1 and we decided to not run any control experiment. Already in the revised version we acknowledged this aspect as a limit of the paper, and we proposed a “what next?” experiment for future work (page 21, lines 933-938). Thus, we decided to specify this aspect in the general discussion, on the base of the positive evaluation obtained in the first round by the reviewers regarding a possible contribution of our study in the field. Moreover, in our opinion, this paper could conclude a set of studies about the mental timeline, given that we published a study for the horizontal mental time (with a procedure very similar to that adopted here), a study about the vertical mental timeline, and a study with the combination of horizontal and vertical information. Thus, the present study could conclude these attempts to contribute to the representation of time (see reviewer#2 in this round).
Reviewer 2 Report
Comments and Suggestions for Authors
The authors were very responsive and have adequately addressed the many points highlighted in the last revision. I want to thank them for their patience, also taking into account the long list of suggestions provided in the first round. I have carefully checked the revised manuscript again and believe the revisions have improved its strength. This paper will be an informative contribution to the literature on the representation of time.
As a sidenote, it would have facilitated the review process if the authors had indicated where exactly the changes could be found. (I acknowledge that I missed some of the details that were indeed provided already in the first version, like the number of trials.) nFinally, I would like to make two last comments:
- The between-participant heterogeneity is now described in terms of the proportion of significant effects. Given that test power is also affected by sample size, I would recommend reporting the range or SD so that readers have an impression of how meaningful these effects are.
- I still feel a bit uncomfortable how the relations between mean RT differences (dRT) with the specific order sequence are currently handled. The beta coefficient mentioned in the text now suggests a linear relation of a dimensional variable, whereas potentially discrepant information has been removed now. Nevertheless, I appreciate that the authors have added a discussion that effects could be categorical or linear, depending on task framing.
Author Response
Reviewer#2 (Reviewer#3 in the first round):
-“The authors were very responsive and have adequately addressed the many points highlighted in the last revision. I want to thank them for their patience, also taking into account the long list of suggestions provided in the first round. I have carefully checked the revised manuscript again and believe the revisions have improved its strength. This paper will be an informative contribution to the literature on the representation of time”.
We would like to thank the reviewer#2 (reviewer#3 of the first round) for her/his positive comments about our effort to improve the quality of the manuscript taking into account all suggestions provided by all reviewers of the first round.
-“As a sidenote, it would have facilitated the review process if the authors had indicated where exactly the changes could be found. (I acknowledge that I missed some of the details that were indeed provided already in the first version, like the number of trials.)”.
We apology whether we did not help the reviewers to easily identify the changes provided in the text. It is our fault, and we are sorry for that. The main reason for the lack of specific indications of changes in the response letter was grounded on the attempt to respect the scheduled deadline for the submission after we asked for an extension of the deadline provided by the Editor and Assistant Editor of the Journal.
-“Finally, I would like to make two last comments:- The between-participant heterogeneity is now described in terms of the proportion of significant effects. Given that test power is also affected by sample size, I would recommend reporting the range or SD so that readers have an impression of how meaningful these effects are”.
Honestly, we did not understand this comment because in the relative tables regarding participant heterogeneity (Table 2 page 10 and Table 4 page 14) the SD has been reported. We only observed that we did not report information about the chi-squared test performed (see page 9 line 404, page 14 line 558, and page 16 line 672). Thus, we decided to add this missing information in the corresponding part of the text.
-“I still feel a bit uncomfortable how the relations between mean RT differences (dRT) with the specific order sequence are currently handled. The beta coefficient mentioned in the text now suggests a linear relation of a dimensional variable, whereas potentially discrepant information has been removed now. Nevertheless, I appreciate that the authors have added a discussion that effects could be categorical or linear, depending on task framing”.
We are proud for the fact that the reviewer#2 (past reviewer#3) appreciated our attempt to discuss the categorical-continuous fashion of the mental timeline according to the nature of the task. In order to help the reviewer#2 to feel a little bit comfortable with the RT difference according to the order reported in the Time-to-Position task, we decided to perform the same analysis described by Beracci et al. (2022; Psychological Research), and the results are reported in the Supplementary Materials, given that the categorical-continuous representation of time was not one of the aims of the present paper. In the Supplementary Materials, we also re-reported the regression figures to reply to the point raised by the reviewer#1.
Round 2
Reviewer 1 Report
Comments and Suggestions for Authors
The authors have made a major effort to answer my critique (I am ref 1) and the paper is now in much better shape, being easier to follow.
Two issues remain. The first is that the authors removed Fig 7 and only provide a regression analysis. Without Fig. 7, however, it is impossible for the reader to tell whether a progressive changes, as implied by linear regression, or a binary split, is a better description of the data.
Just looking at Fg 7 in the first draft, it was obvious to me (and to ref 3) that a binary split is the better description. To show this, I suggested calculating a root mean square (RMS) indication of fit. For example, calculate the square-root of the sum of the squared deviations between (a) all the data and the best-fit straight line (i.e., the regression), for comparison with (b) the square-root of the sum of the squared deviations between all the data below the split and its mean value plus all the squared deviations between the data above the split and its mean value, making sure to use the same code so as to have truly equivalent values. I will be shocked if the RMS for linear regression is smaller than that for the binary split. (Since each analysis has two free parameters (mean and slope for regression, two means for the binary split), and applies to the same number of data points, there is no need to correct for the degrees of freedom, which will be equal.) At all events, there is no reason to remove the graph; it should stay, ideally with both types of predictions (regression and binary split) shown with the data.
The more basic issue is, however, that the authors have apparently not run a control for the typical direction of scrolling, so they have not shown that the results have anything to do with a mental time line rather than a motor habit. I had suggested running the experiment on old people, but most likely this will not work as everyone has been scrolling nowadays, old and young. So how about using non-temporal information, such as type of font, and testing the differences between reaction times for upwards and downwards scrolls ? The authors need to think about a suitable control for the motor habit, before I would be positive for publication. I hope that they do, because otherwise this is a fine piece of work and could be a real contribution to the field.
Comments on the Quality of English LanguageNumerous small edits are needed to remove italian-style phrases,
but the content is nevertheless clear.
Author Response
Reviewer#1 second round:
“Numerous small edits are needed to remove italian-style phrases, but the content is nevertheless clear”.
We thank the reviewer#1 for this suggestion. A English native-speaker checked all phrases and modified them to reduce Italian-style phrases. We hope that now the paper is suitable for the publication.